# Resource Carrying Capacity Evaluation Based on Fuzzy Evaluation: Validation Using Karst Landscape Region in Southwest China

Xinhao Min [1], Yanning Wang [1,2,*] and Jun Chen [3]

[1] MOE Key Laboratory of Intelligence Manufacturing Technology, Department of Civil and Environmental Engineering, Shantou University, Shantou 515063, China
[2] State Key Laboratory for Geomechanics and Deep Underground Engineering, China University of Mining and Technology, Xuzhou 221000, China
[3] Department of Civil Engineering, Shanghai Jiao Tong University, Shanghai 200030, China
[*] Correspondence: wangyn@stu.edu.cn; Tel.:+13642206669

**Abstract:** The problems of regional resource shortage, fragile ecological environment and unbalanced social development are becoming increasingly serious. There is an urgent need for rational evaluation and planning of resources and the environment. This paper presents a fuzzy comprehensive evaluation method combined with Analytic Hierarchy Process (AHP) and shortcoming element evaluation to analyze the resource and environmental bearing capacity of a certain region. The proposed model was verified by backing data analysis from a karst landscape region in southwest China. Short board element analysis was employed for further study. The results show that (a) the calculation results of the evaluation system used in this paper are consistent with the actual situation. The method can be effectively used in the field of resource and environmental carrying capacity evaluation. (b) The environmental carrying capacity is the largest in this region, followed by the resource carrying capacity, and the socio-economic carrying capacity is the smallest. (c) The region has a sufficient environmental carrying capacity on the whole, the resource conditions are weak and the socio-economic development is backward. The analysis of the evaluation results provides a scientific basis for the rational use of resources, territorial spatial planning, sustainable socio-economic development and ecological environmental protection strategies in karst mountainous areas.

**Keywords:** fuzzy comprehensive evaluation method; Analytic Hierarchy Process; short board elements; environmental carrying capacity

## 1. Introduction

Bearing force, originally a mechanical concept, dates back to Malthusian times. In 1798, Malthus published his work *The Principle of Population*, which gave a modern connotation to carrying capacity. It has had a significant and far-reaching impact on the study of the concepts of carrying capacity, biology, ecology and demography [1]. There is a variety of ecosystems, large and small, in the world, each with its own unique carrying capacity, as shown in Figure 1.

With the rapid development of the economy and society, the problems of resources and the environment are becoming increasingly serious. How to scientifically and quantitatively study sustainable development among resources, environment and economy is gradually becoming a common concern in academic circles [2,3]. Especially in developing countries, environmental pollution and irrational exploitation of resources have become problems that cannot be ignored [4].

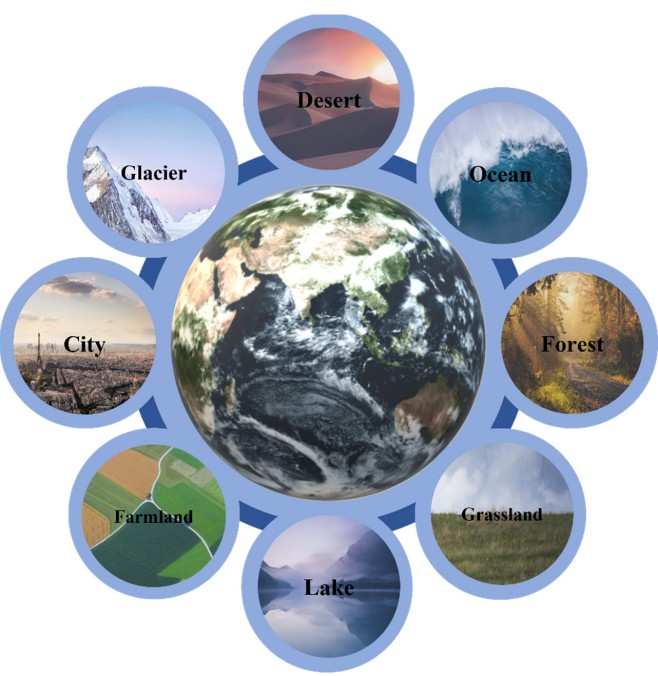

**Figure 1.** Ecosystems.

The evaluation of the bearing capacity of resources and the environment has grown from a single study on the bearing capacity of land resources and water resources to a comprehensive evaluation [5–10]. Scholars from various countries have established different research systems according to different research fields. Wang and Zeng proposed an Inexact Fuzzy Multi-Objective Planning model (IFMOP) based on environmental carrying capacity for the industrial structure optimization problem [11]. However, the IFMOP model requires a large amount of data to determine its parameter distribution and cannot integrate uncertain information into the optimization results. Chen built a model unifying all the indicators into one to calculate the final bearing capacity level by using the Tourism Ecological Footprint (TEF) and Tourism Ecological Capacity (TEC) and evaluated the ecological status of Golden Gate National Park from 2002 to 2011 [12]. However, for the bearing capacity calculation of multiple indicators, it is difficult for this method to determine the transformation between indicators. Wang et al. established an environmental early warning indicator system based on the Driver–Pressure–State–Impact–Response (DPSIR) model. Analytic Hierarchy Process (AHP) was used to determine the weights. Next, the single-indicator method and the integrated indicator method were used to further evaluate the environmental Early Warning Status, in which the weighted summation method was used to summarize the data and results [13]. However, the traditional DPSIR model aims to describe the changes in lifestyle, production and consumption patterns brought about by socio-economic development and population growth, and therefore focuses more on non-environmental factors and neglects ecological factors. This paper uses a fuzzy comprehensive evaluation method and hierarchical analysis method combined with short board elements to calculate representative indicators of resource and environmental carrying capacity. The fuzzy evaluation method can solve the fuzzy or difficult quantification problems. Resource and environmental carrying capacity evaluation is essentially an analysis of the combination of definite evaluation indexes and their uncertain evaluation factors, for example, the degree of construction land development or the degree of arable land utilization in the resource carrying capacity. These issues are difficult to analyze quantitatively. The application of fuzzy comprehensive analysis can better solve this problem [14,15].

In past studies, there were limitations and shortcomings, along with rich results. Most of the research scales are cross-provincial [16–19] and municipal [20–22] studies; there are not many studies on the evaluation of carrying capacity at the county scale [23]. In terms

of research regions in China, current studies mostly focus on the less developed western regions [24,25], water source ecological regions [26,27], mountainous and hilly regions [28,29], eastern coastal regions [30,31] and urban economic clusters [32,33]. However, there are fewer resource and environmental carrying capacity evaluations for mountainous areas with complex topography, rich tourism resources and urgent agricultural problems in karst landscapes [34]. As a county-level administrative region in southwestern China, Shilin County in the karst mountainous region suffers from scarce resources, a fragile ecological environment and backward socio-economic development, and lacks studies about its resource carrying capacity. The present study fills this research gap. Understanding the resource and environmental carrying capacity of Shilin County can be beneficial for the rational development and utilization of resources, ecological and environmental protection, and sustainable socio-economic development [35].

In view of this, this paper takes Shilin County, Yunnan Province, as an example, and uses the county level as the evaluation unit scale. The index system is established according to three subsystems: water and soil resource carrying capacity, ecological and environmental carrying capacity, and social and economic carrying capacity. Firstly, the Analytic Hierarchy Process was used to assign comprehensive weights. Secondly, the fuzzy comprehensive evaluation method was used to calculate the comprehensive score of the resource and environmental carrying capacity of Shilin County in 2020. Finally, combined with the analysis of shortcoming elements, the enhancement strategy is proposed. It provides a reference basis for the future resource and environmental planning of karst areas at home and abroad. The flowchart of the comprehensive evaluation method is shown in Figure 2.

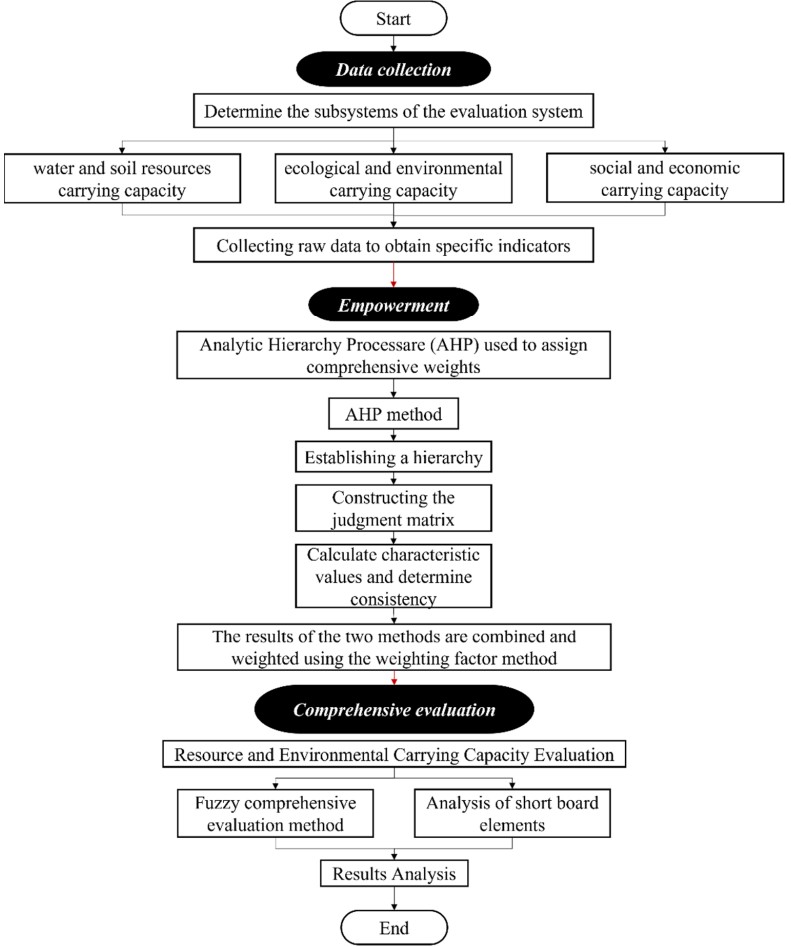

**Figure 2.** Flowchart of comprehensive evaluation method.

## 2. Research Area Overview and Data Acquisition

Shilin Yizu Autonomous County (Shilin County) is located in a typical highland karst mountainous area in southwestern China and southeastern Yunnan Province. The geographical location of Shilin County is shown in Figure 3.

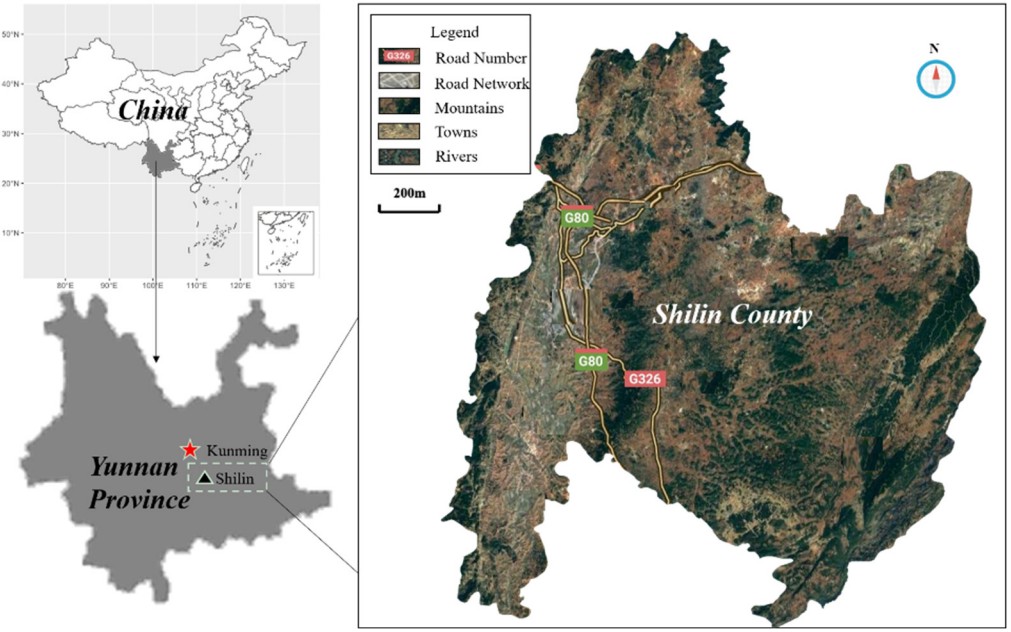

**Figure 3.** Location of Shilin County. Satellite image from www.gisrs.cn (accessed on 2 April 2022).

It is located at 24°46′27.55″ N, 103°17′18.83″ E. The county is 57.3 km wide from east to west and 58.8 km long from north to south, with a total land area of 1719 square kilometers and an average altitude of 1688 m [36]. The topography of Shilin County is complex, with gently undulating terrain that slopes down in steps from east to west. The mountains and rivers basically stretch from north to south. The county is divided into three main types of landforms: karst landforms, mountains and basins. Karst is the most characteristic geographical landscape in Shilin County; two-thirds of the county are karst landforms, mainly distributed in the central part of the county [37]. The average annual rainfall in the county is 967.9 mm. The dry period is from December to April every year; the semi-wet period is in May, October and November; and the wet period is from June to September. The average annual evaporation in the country is 2097.7 mm, with the maximum value occurring in April, at 321.1 mm. The minimum value is in November, at 105.5 mm [38]. The classification of precipitation levels is shown in Table 1.

**Table 1.** Annual precipitation level classification table (data from http://www.gov.cn/ (accessed on 2 April 2022)).

| Precipitation Level | Annual Precipitation (mm) | Features |
|---|---|---|
| Wet period | >800 | The air is moist and evaporation is low. |
| Semi-wet period | 400–800 | Both precipitation and evaporation are high. |
| Dry period | 200–400 | The evaporation significantly exceeds the precipitation. |
| Semi-dry period | <200 | Precipitation is low and evaporation is high. |

The county is rich in mineral resources due to its special geological formations and stratigraphic development. There are more than 10 kinds of mineral resources, including coal, copper, iron, silver, phosphorus, lead, zinc, limestone, marble, sulfur, cadmium, cobalt,

oil shale, quartz sand, alum, etc., that have been proven. Among the advantageous resources are mainly limestone, coal, lead, zinc, iron, phosphorus, copper, etc. Shilin County is also rich in biological resources. Its unique topography and climate and colorful vegetation distribution are conducive to the reproduction and growth of animal populations [39]. The results of continuous automatic monitoring indicators of the atmosphere in Shilin County in 2021 showed that the excellent air quality rate reached 99.7%, indicating a good regional atmospheric environment. In 2021, the total population of the county was 240,800, the urbanization rate was 40.1% and the per capita GDP was 44,013.44 yuan [40]. In recent years, Shilin's resources have faced massive depletion and unreasonable exploitation. This has caused many negative impacts such as land degradation, shortage of water and soil resources, ecological environment deterioration and backward economic and social development. It has put great pressure on regional resources and the environment.

The water resources data are mainly from the 13th Five-Year Plan and the 2021 Water Resources Bulletin of Shilin Yizu Autonomous County. Land resources data are from Shilin Yizu Autonomous County People's Government on the Publication of Shilin Yizu Autonomous County Land Use Master Plan (2010–2020) [41]. The occupancy of land resource use types is shown in Figure 4 and Table 2.

Mineral resources data are from the Annual Report on Statistical Information of Natural Resources Management in Shilin Yizu Autonomous County (2021) [42]. Atmospheric environmental data were obtained from the continuous automatic atmospheric monitoring information of Shilin County Ecological Environment Bureau [43]. Socio-economic situation data are from the Statistical Bulletin of National Economic and Social Development of Shilin Yizu Autonomous County in 2021 [44]. The population data were obtained from the Results of the Seventh National Population Census [45].

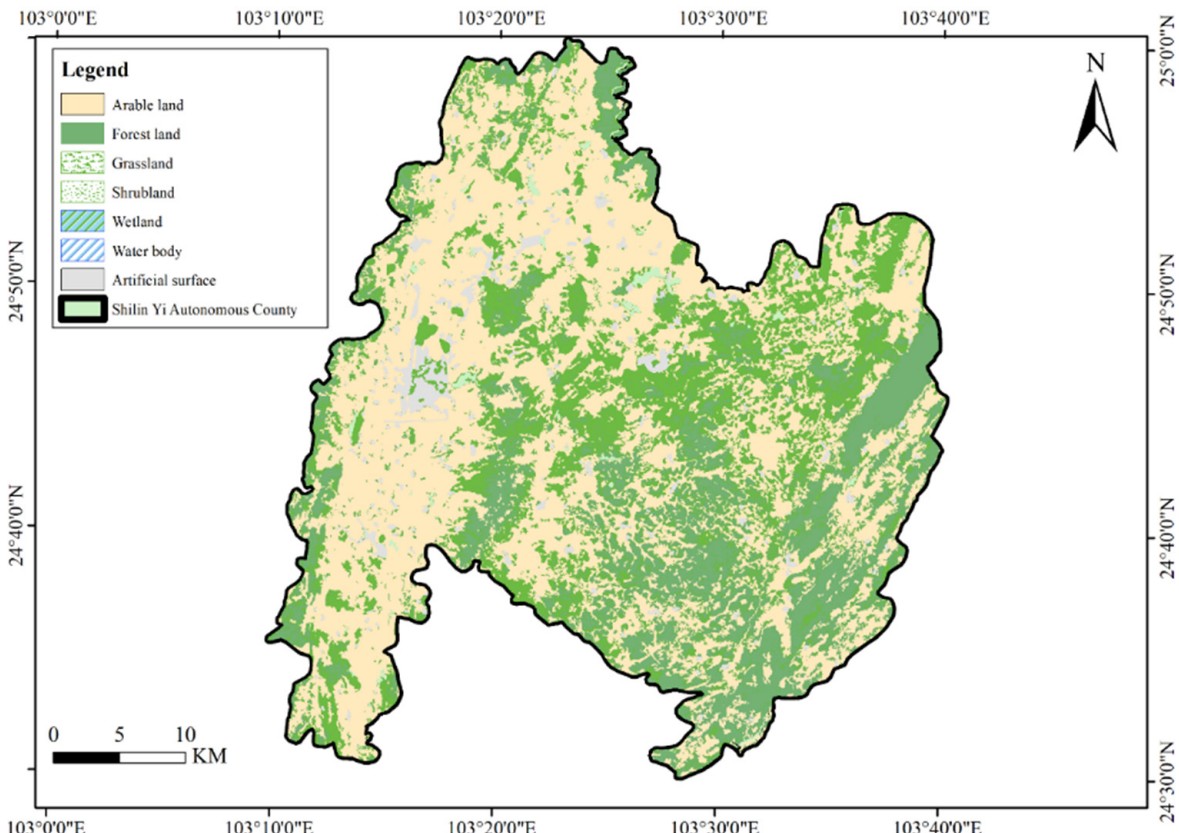

**Figure 4.** Shilin County land use type coverage map.

**Table 2.** Shilin County land use types.

| Land Use Type | | Land Area (Million Hectares) | |
|---|---|---|---|
| Arable land | Water field | 0.5964 | |
| | Watered land | 0.1963 | 5.6672 |
| | Dryland | 4.8745 | |
| Forest land | Tree woodland | 6.3816 | |
| | Bamboo woodland | 0.0149 | 7.5796 |
| | Other forest land | 1.1831 | |
| Wetland | Marshland | 0.4752 | |
| | Mudflat | 0.7218 | 1.2 |
| | River | 0.0177 | |
| | Lake | 0.0009 | |
| Water body | Pond | 0.0897 | 0.2724 |
| | Reservoir | 0.1387 | |
| | Ditch | 0.0254 | |
| Shrubland | | 0.987 | |
| Artificial surface | | 1.0688 | |
| Grassland | | 0.2022 | |

## 3. Resource Carrying Capacity Evaluation Model Construction

### 3.1. Construction of Evaluation Index System

The establishment of the index system for the bearing capacity of resources and the environment in karst areas should be based on the principles of following scientificity and comprehensiveness. Then, on this basis, the indicators are determined by combining the availability of indicators and the natural environment and socio-economic characteristics of karst areas. The paper draws on relevant literature and technical regulations, taking the resource and environmental carrying capacity of karst areas as the target layer and dividing three guideline layers and 27 specific indicators, as shown in Figure 5. For details, refer to Tables 3 and 4.

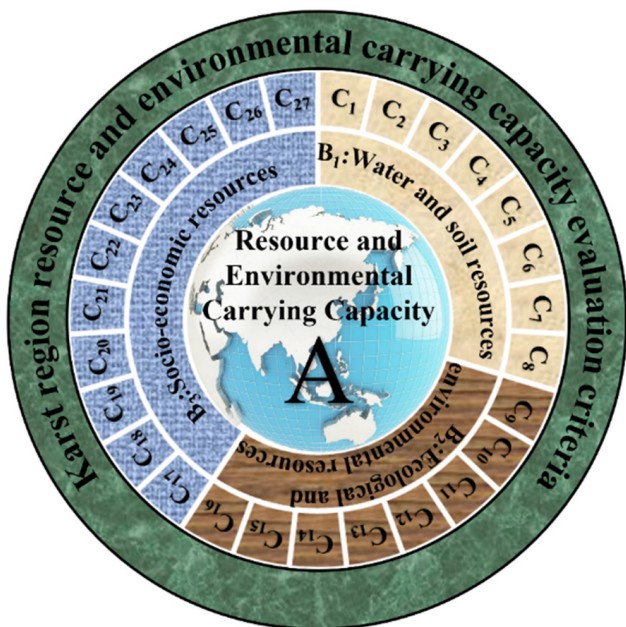

**Figure 5.** Karst region resource and environmental carrying capacity evaluation index map.

**Table 3.** Karst region resource and environmental carrying capacity evaluation criteria.

| Dimensions | Designator | Indicator |
|---|---|---|
| Target Level | A | Karst region resource and environmental carrying capacity |
| Guideline level | B1 | Water and soil resources carrying capacity |
| | B2 | Ecological and environmental resource carrying capacity |
| | B3 | Socio-economic resource carrying capacity |
| Indicator Level | C1 | Arable land per capita |
| | C2 | Arable land development and utilization degree |
| | C3 | Construction land area per capita |
| | C4 | Current development level of construction land |
| | C5 | Water resources per capita |
| | C6 | Grain production per capita |
| | C7 | Building area to arable land ratio |
| | C8 | Land utilization rate |
| | C9 | Forest coverage |
| | C10 | Greening coverage of built-up areas |
| | C11 | Air quality attainment rate for days at level 2 and above |
| | C12 | Ground cover rate |
| | C13 | Fertilizer use per unit area |
| | C14 | Pesticide use per unit area |
| | C15 | Percentage of ecological area |
| | C16 | Percentage of wind-sanded land area |
| | C17 | GDP per capita |
| | C18 | Economic density |
| | C19 | Population density |
| | C20 | Agricultural machinery power per capita |
| | C21 | Arable land production capacity per capita |
| | C22 | Per capita net income of farmers |
| | C23 | Energy consumption of 10,000 yuan GDP |
| | C24 | Electricity consumption of 10,000 yuan GDP |
| | C25 | GDP per unit of construction land |
| | C26 | Environmental investment as a percentage of GDP |
| | C27 | The proportion of total output value of secondary and tertiary industries |

**Table 4.** Karst region resource and environmental carrying capacity index system.

| Target Level | Guideline Level | Indicator Level | Indicator Unit | Indicator Attribute |
|---|---|---|---|---|
| A | B1 | C1 | $hm^2$/person | + |
| | | C2 | - | + |
| | | C3 | $hm^2$/person | + |
| | | C4 | - | + |
| | | C5 | million tons/person | + |
| | | C6 | million tons/person | + |
| | | C7 | % | + |
| | | C8 | % | + |
| | B2 | C9 | % | + |
| | | C10 | % | + |
| | | C11 | % | + |
| | | C12 | $t/hm^2$ | - |
| | | C13 | $t/hm^2$ | - |
| | | C14 | % | - |
| | | C15 | % | + |
| | | C16 | - | - |
| | B3 | C17 | billion yuan/person | + |
| | | C18 | million yuan/$hm^2$ | + |
| | | C19 | person/$hm^2$ | - |
| | | C20 | kW/person | + |
| | | C21 | million tons/person | + |
| | | C22 | million yuan | + |
| | | C23 | tce/million yuan "tce" is "ton of standard coal equivalent" | - |
| | | C24 | million kW·h/million yuan | - |
| | | C25 | billion yuan/$hm^2$ | + |
| | | C26 | % | + |
| | | C27 | % | + |

The key indicators are calculated as follows.

(1)   The degree of arable land exploitation is used to analyze the relationship between current arable land and arable land reserve resources. The formula is as follows.

$$\text{Arable land development and utilization degree } (C_2) = \frac{\text{Current total arable land area}}{(\text{Current total arable land area} + \text{Arable land reserve area})} \quad (1)$$

(2)   Current development level of construction land.

The current development level of construction land is used to analyze the relationship between the development intensity and the ultimate development intensity. The formula is as follows.

$$\text{Current development level of construction land } (C_4) = \frac{\text{Current development intensity}}{\text{Extreme development intensity}} \quad (2)$$

In Equation (2), the current development intensity is **DI** and the extreme development intensity is **LDI**.

Based on the evaluation results of the suitability of construction and development, the two categories of the most suitable space and the basically suitable space and the space of the current construction land are considered as the limit development scale. The limit development intensity and the current development intensity are measured. The formula is as follows.

$$LDI = \frac{(E_1 + E_2)C}{S} \quad (3)$$

$$DI = \frac{C}{S} \quad (4)$$

In Equations (3) and (4), **C** is the area of the current construction land in region and **S** is the total area of the country of the evaluation unit. $E_1$ and $E_2$ are the most suitable and basically suitable areas, respectively, in the evaluation of the suitability of land construction and development.

(3)   Agricultural machinery power per capita.

Agricultural machinery power per capita refers to the ratio of total regional agricultural machinery power to agricultural population. This is an indicator that reflects the regional agricultural production capacity. The formula is as follows.

$$\text{Agricultural machinery power per capita } (C_{20}) = \frac{\text{Total regional agricultural machinery power}}{\text{Total regional agricultural population}} \quad (5)$$

(4)   Arable land production capacity per capita.

The production capacity of arable land per capita refers to the amount of crops produced on the arable land acquired per person in the region, and is an indicator of the production capacity of arable land in the region. The formula is as follows.

$$\text{Arable land production capacity per capita } (C_{21}) = \frac{\text{Total crop production}}{\text{Total regional population}} \quad (6)$$

*3.2. Standardization of Evaluation Indicators*

The difference in magnitudes among the indicators makes it impossible to use the raw data directly for bearing capacity evaluation. Therefore, standardization is needed to eliminate the difference in magnitudes. The paper uses a logarithmic pattern normalization method to convert logarithmic functions with a base of 10. The expressions of the logarithmic function conversion are shown below.

$$y = log_{10}(x) \quad (7)$$

$$\text{Standardized results} = \frac{1}{1 + e^{(-\text{ Original data})}} \quad (8)$$

### 3.3. Determination of Evaluation Index Weights

In this paper, the Analytic Hierarchy Process (AHP) is used to assign weights to ensure the reasonableness and accuracy of index weights. AHP decomposes the decision problem into different hierarchical structures in the order of general objectives, sub-objectives at each level, and evaluation criteria up to specific alternative investment options. Then, the weights of the relative order of importance of the elements in the same level are determined, and the hierarchy is sorted. Finally, the sorting results are analyzed [46–48].

The sum method in the AHP is used. The following are the calculation steps.

(1) Establish a hierarchy.
(2) Construct the judgment matrix.

The judgment matrix represents the relative importance between the elements $\mathbf{B_1}$, $\mathbf{B_2}$, ... ; $\mathbf{B_n}$ related to an element $\mathbf{A}$ at that level for the previous level. Details are shown in Table 5.

**Table 5.** Judgment matrix.

| A | $\mathbf{B_1}$ | $\mathbf{B_2}$ | $\cdots$ | $\mathbf{B_n}$ |
|---|---|---|---|---|
| $B_1$ | $b_{11}$ | $b_{12}$ | $\cdots$ | $b_{1n}$ |
| $B_2$ | $b_{21}$ | $b_{22}$ | $\cdots$ | $b_{2n}$ |
| $\vdots$ | $\vdots$ | $\vdots$ | | $\vdots$ |
| $B_n$ | $b_{n1}$ | $b_{n2}$ | $\cdots$ | $b_{nn}$ |

The relative importance of factors $b_i$ and $b_j$ is usually taken as values of 1, 3, 5, 7, 9 and their reciprocals.

The evaluation factors ($C_1$, $C_2$, ... , $C_n$) of each indicator layer in Table 3 are brought to the corresponding positions ($B_1$, $B_2$, ... , $B_n$) in the judgment matrix in Table 5. Teachers from the Department of Civil and Environmental Engineering of Shantou University were invited to score the evaluation factors.

(3) Calculate characteristic values and determine consistency.

First, each column vector of the judgment matrix $\mathbf{A}$ is normalized.

$$\widetilde{w}_{ij} = \frac{b_{ij}}{\sum_{i=1}^{n} b_{ij}} (i, j = 1, 2, 3, \cdots, n) \tag{9}$$

Second, the normalized rows are summed.

$$\widetilde{w}_i = \sum_{j=1}^{n} \widetilde{w}_{ij} (i, j = 1, 2, 3, \cdots, n) \tag{10}$$

Then, the row vectors are normalized.

$$w_i = \frac{\widetilde{w}_i}{\sum\limits_{i=1}^{n} \widetilde{w}_i} (i = 1, 2, 3, \cdots, n) \tag{11}$$

$$\overrightarrow{W} = (w_1, w_2, \cdots, w_n)^T (i = 1, 2, 3, \cdots, n) \tag{12}$$

Finally, the approximation of the maximum characteristic value is calculated to determine the consistency.

$$\lambda_{\max} = \frac{1}{n} \sum_{i=1}^{n} \frac{\left(\overrightarrow{A}\overrightarrow{W}\right)_i}{w_i} (i = 1, 2, 3, \cdots, n) \tag{13}$$

$$CI = \frac{\lambda_{\max} - n}{n - 1} \tag{14}$$

$$CR = \frac{CI}{RI} \tag{15}$$

when **CI** = 0, the judgment matrix is completely consistent. The consistency ratio **CR** was obtained using **CI** with the average random consistency index **RI**. If **CR < 0.1**, the consistency of the judgment matrix meets the study requirements. Otherwise, the relative importance among indicators needs to be redetermined. The average random consistency index **RI** is shown in Table 6 [47–49].

**Table 6.** Average random consistency index.

| *n* | 1 | 2 | 3 | 4 | 5 | 6 | 7 | 8 | 9 | 10 | 11 |
|-----|---|---|------|------|------|------|------|------|------|------|------|
| **RI** | 0 | 0 | 0.52 | 0.89 | 1.12 | 1.26 | 1.36 | 1.41 | 1.46 | 1.49 | 1.52 |

*3.4. Comprehensive Evaluation*

The evaluation value of each index is considered comprehensively, and the fuzzy evaluation method is used to calculate the comprehensive score to eliminate the influence of the singularity of the index. The fuzzy evaluation method makes a comprehensive evaluation of the bearing capacity of resources and the environment from multiple levels, which can consider many influencing factors and help avoid the problem of deviation from the objective reality. Therefore, the spatial scale is taken as the basic scale of resource and environmental bearing capacity measurement, the fuzzy comprehensive evaluation method is used to calculate the level characteristic value (**T**), the affiliation degree of spatial parcels in each level is calculated by the descending semi-trapezoidal distribution function, and the integrated affiliation result is obtained as the comprehensive evaluation result of resource and environmental bearing capacity. The specific operation of the model is as follows [50].

(1) According to the evaluation index system of resource and environmental carrying capacity and the grading threshold of each index, the evaluation index **X** and the evaluation ensemble **Y** are set.

$$X = \{X_1, X_2, \cdots, X_n\}, \ Y = \{Y_1, Y_2, \cdots, Y_m\}$$

when *j* = 1,

$$y_{ij} = \begin{cases} 1 & x_i \le s_{ij} \\ \frac{x_i - s_{i(j+1)}}{s_{ij} - s_{i(j+1)}} & s_{ij} < x_i \le s_{i(j+1)} \\ 0 & x_i > s_{i(j+1)} \end{cases} \tag{16}$$

when *j* = 2, 3, $\cdots$, *m* − 1,

$$\begin{cases} 1 & x_i \le s_{i(j-1)} \\ \frac{x_i - s_{i(j-1)}}{s_{ij} - s_{i(j-1)}} & s_{i(j-1)} < x_i \le s_{ij} \\ 1 & x_i = s_{ij} \\ \frac{x_i - s_{i(j+1)}}{s_{ij} - s_{i(j+1)}} & s_{ij} < x_i \le s_{i(j+1)} \\ 0 & x_i > s_{i(j+1)} \end{cases} \tag{17}$$

when *j* = *m*,

$$y_{ij} = \begin{cases} 1 & x_i > s_{ij} \\ \frac{x_i - s_{i(j-1)}}{s_{ij} - s_{i(j-1)}} & s_{i(j-1)} < x_i \le s_{ij} \\ 0 & x_i \le s_{i(j-1)} \end{cases} \tag{18}$$

where $y_{ij}$ is the affiliation degree of rank *j* with indicator *i*. $x_i$ is the actual value of the indicator. $s_{ij}$ is the grading threshold for a grade *j* with indicator *i*.

The resource carrying capacity score interval is 0–1. In order to make the evaluation results more scientific and reasonable, the evaluation score is divided into five grades according to the statistical principle. The grade division is shown in Table 7.

**Table 7.** Resource and environmental carrying capacity evaluation level.

| Evaluation Level | Excellent | Good | Medium | Poor | Terrible |
|---|---|---|---|---|---|
| Range of Values (T) | 0.8~1 | 0.6~0.8 | 0.4~0.6 | 0.2~0.4 | 0~0.2 |

(2) The fuzzy matrix weighting is applied to calculate the total affiliation of each subsystem of bearing capacity and the integrated bearing capacity under each level. To ensure that the information is complete and accurate, the total affiliation degree is calculated by the weighted evaluation fuzzy integrated operator method.

$$D_j = \sum_1^n (d_i)_{1\times 5} = \sum_1^n (w_i \times y_{ij})_{1\times 5} \tag{19}$$

where $D_j$ is the total affiliation degree with bearing capacity class $j$, and $d_j$ is the affiliation degree of the $i$ factor to each level of the evaluation index. It represents the degree of association of the indicator with each factor. $n$ is the number of evaluation indicators, $w_i$ is the weight of indicator $i$ and $y_{ij}$ is the affiliation degree of rank $j$ with indicator $i$.

(3) Calculate the level characteristic values of each subsystem of the bearing capacity and the integrated bearing capacity. The level affiliation is the affiliation of each sample to each level fuzzy subset, which is a fuzzy vector rather than a point value, and although it provides more complete information, it is not easy to express the comprehensive level of the sample. The total level characteristic value (*T*) is calculated by weighted average summation, which is used as the evaluation index of the regional resource and environmental carrying capacity.

$$T = \sum_{j=1}^5 (d_i \times j)_{1\times 5} = \frac{\sum\limits_{j=1}^5 D_j \times j}{\sum\limits_{j=1}^5 D_j} \tag{20}$$

(4) Analysis of short board elements.

Based on a full understanding of the carrying capacity of Shilin County's resources and environment and the carrying capacity scores of its various subsystems, representative indicators are selected and shortcomings are analyzed, taking into account actual needs and data availability, in order to propose targeted strategies.

The current state of each indicator in 2021 is compared with the threshold interval, and the state index (*R*) of the positive and negative indicators is calculated using the extreme difference normalization method to measure the state level at which the indicator is located. The state index (*R*) is calculated by the following formula.

$$R = \frac{V_{current} - V_{min}}{V_{max} - V_{min}} \tag{21}$$

where $V_{current}$ is the actual state value of the indicator, $V_{max}$ is the maximum value of the indicator in the threshold interval and $V_{min}$ is the minimum value of the indicator in the threshold interval.

## 4. Calculation of the Resource Carrying Capacity of Shilin County and Analysis of the Shortage Elements

### 4.1. Calculation of the Resource Carrying Capacity of Shilin County

According to the results information and index calculation formula, the raw data of each index were obtained by quantitative calculation. The raw data were standardized by using the logarithmic model standardization method for evaluation and analysis.

First, the index weights were calculated using hierarchical analysis. The relative weights of the indicator layers included in the three guideline layers were calculated as shown in Figure 6.

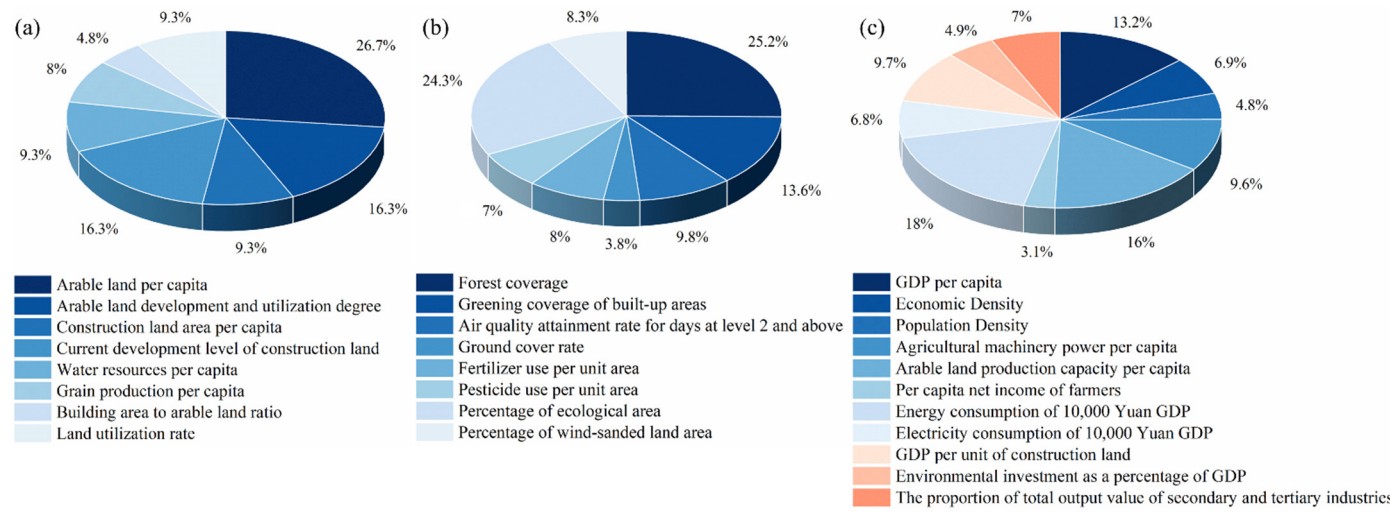

**Figure 6.** Indicator weights for the three orientation layers. (**a**) Water and soil resources carrying capacity indicator layer. (**b**) Ecological environment carrying capacity indicator layer. (**c**) Socio-economic carrying indicator layer.

Based on the combined weights of each indicator, the relative weights among the indicators in the subsystem were calculated. The fuzzy comprehensive evaluation of the carrying capacity of three subsystems in Shilin County is based on Equations (16)–(20). The calculation results are shown in Table 8.

**Table 8.** Shilin County resource and environmental carrying capacity evaluation table.

| Guideline Level | Indicator Level | Original Data | Standardized Value | Relative Weights | Indicator Score | Load-Bearing Capacity Score |
|---|---|---|---|---|---|---|
| B1 | C1 | 0.2353 | 0.5586 | 0.2673 | 0.1493 | 0.6074 |
| | C2 | 0.9806 | 0.7272 | 0.1633 | 0.1188 | |
| | C3 | 0.0252 | 0.5063 | 0.0927 | 0.0469 | |
| | C4 | 0.5584 | 0.6361 | 0.1631 | 0.1037 | |
| | C5 | 0.0988 | 0.5247 | 0.0927 | 0.0486 | |
| | C6 | 0.0006 | 0.5001 | 0.0804 | 0.0402 | |
| | C7 | 75.03 | 0.6792 | 0.0478 | 0.0325 | |
| | C8 | 97.71 | 0.7265 | 0.0927 | 0.0673 | |
| B2 | C9 | 43.36 | 0.6067 | 0.2521 | 0.1529 | 0.6038 |
| | C10 | 39.37 | 0.5972 | 0.1359 | 0.0812 | |
| | C11 | 99.7 | 0.7305 | 0.0982 | 0.0717 | |
| | C12 | 8.27 | 0.5207 | 0.0375 | 0.0195 | |
| | C13 | 0.6352 | 0.6537 | 0.0799 | 0.0522 | |
| | C14 | 0.0082 | 0.5020 | 0.0709 | 0.0356 | |
| | C15 | 46.33 | 0.6138 | 0.2428 | 0.1490 | |
| | C16 | 12.3 | 0.5031 | 0.0827 | 0.0416 | |

**Table 8.** *Cont.*

| Guideline Level | Indicator Level | Original Data | Standardized Value | Relative Weights | Indicator Score | Load-Bearing Capacity Score |
|---|---|---|---|---|---|---|
| B3 | C17 | 0.0003 | 0.5001 | 0.1323 | 0.0662 | 0.5665 |
| | C18 | 6.8104 | 0.5170 | 0.0691 | 0.0357 | |
| | C19 | 140.0814 | 0.8023 | 0.0479 | 0.0384 | |
| | C20 | 4.3956 | 0.5110 | 0.0962 | 0.0492 | |
| | C21 | 0.0006 | 0.5001 | 0.1599 | 0.0800 | |
| | C22 | 1.7440 | 0.5044 | 0.0312 | 0.0157 | |
| | C23 | 0.8201 | 0.6943 | 0.1798 | 0.1248 | |
| | C24 | 0.1007 | 0.5252 | 0.0682 | 0.0358 | |
| | C25 | 0.0156 | 0.5039 | 0.0966 | 0.0487 | |
| | C26 | 1.96 | 0.5049 | 0.0488 | 0.0246 | |
| | C27 | 73.8 | 0.6766 | 0.0700 | 0.0474 | |

*4.2. Analysis of Short Board Elements*

The status level indices of the short board elements are shown in Table 9.

**Table 9.** Status index level classification.

| Status Index Range | Status Level |
|---|---|
| R < −1.0 | Crisis Status (Shortage Factors) |
| −1.0 ≤ R < 0.0 | Early Warning Status (Limiting Factors) |
| 0.0 ≤ R < 1.0 | General Status |
| R ≥ 1.0 | Great Status |

Referring to the Urban Land Classification and Planning Construction Land Standard, Village and Township Planning Standard (GB50188-93) and the maximum and minimum values of the corresponding indicators in Yunnan Province and other information, the threshold interval is determined. The status index and level of each indicator are measured according to the status index calculation method. The details are shown in Table 10.

There is no shortage factor in the bearing capacity of resources and the environment in Shilin County in Crisis Status. However, the land utilization rate status index is less than 0, which is in the Early Warning Status. GDP per unit of construction land is close to zero and is on the verge of warning. The per capita water possession and forest coverage are 4.0425 and 4.6433, respectively, and the indicator data are much larger than the maximum value of the threshold interval, which is an over-standard factor.

**Table 10.** Indicator status indices and levels.

| Guideline Level | Indicator Level | Threshold Interval | Indicator Data | Status Index | Status Level |
|---|---|---|---|---|---|
| B1 | C1 | 0.1161–0.1612 | 0.2353 | 2.6430 | Great Status |
| | C3 | 0.0056–0.014 | 0.0252 | 2.3333 | Great Status |
| | C5 | 167–371 | 988 | 4.0245 | Great Status |
| | C8 | 97.97–99.64 | 97.71 | −0.1557 | Early Warning Status |
| B2 | C9 | 1.71–10.68 | 43.36 | 4.6433 | Great Status |
| B3 | C25 | 104.97–172.01 | 156 | 0.7612 | General Status |

**5. Results and Analysis**

*5.1. Analysis of Results*

In previous studies, there has been little work on resource carrying capacity evaluation of karst mountains, which is an area in urgent need of research. In this paper, we use fuzzy evaluation and hierarchical analysis to evaluate the resource carrying capacity of

Shilin County, which is a karst mountainous region. This paper uses a combination of AHP and fuzzy evaluation methods. It can reflect the multi-objective, uncertain and fuzzy characteristics of parameters of complex systems. In this paper, it is applied to the field of resource and environmental carrying capacity. It can directly introduce the uncertainty information of the system in the model construction and objectively reflect the real problems, having significant advantages over other simple models.

(1) A specific method of AHP and the fuzzy evaluation method applied to the calculation of resource and environmental carrying capacity was found. The example study of the resource and environmental bearing capacity of Shilin County shows that this model has suitable applicability.

(2) The calculation model in this paper can reflect the essence of the carrying capacity problem of a region and meet the requirements of regional construction for the development of resource carrying capacity. It serves as a scientific basis for decision making.

(3) Shilin County has a comprehensive score of 0.5969 for resource and environmental carrying capacity, which is at an average level. Among them, the bearing capacity of soil and water resources scored 0.6074, which is at a good level. The ecological environmental carrying capacity score is 0.6038, which is at a good level. The socio-economic carrying capacity score is 0.5665, which is at a medium level. The bearing capacity scores of the three subsystems are, in descending order, the bearing capacity of soil and water resources, the bearing capacity of the ecological environment and the bearing capacity of social economy.

(4) In the soil and water resources carrying capacity subsystem, the arable land area per capita has the largest weight. In 2020, the cultivated land in Shilin County accounted for 32.97% of the total land area, and the degree of development and utilization of cultivated land was 98.06%, a high degree of development. Combined with the analysis of shortage elements, the arable land area per capita and the construction land area per capita are in Great Status. However, the land utilization rate is in Early Warning Status, which shows that the per capita arable reserve resources are insufficient, indicating that the arable land area in Shilin County is influenced by the topography and land desertification. The government announcement [51] of Shilin County states that Shilin will adjust the nature of the land and add a new scale of reserve arable land to ease the arable land tension. This is consistent with the conclusion reached in this paper, which verifies the validity of the model.

(5) In the ecological environmental carrying capacity subsystem, the forest coverage rate has the largest weight. In 2020, the forest coverage rate of Shilin County was 43.36%; combined with the analysis of the shortcoming elements, Shilin County is still in Great Status. However, the percentage of wind-sanded land area is relatively low, at 12.3%. The percentage of ecological land area is relatively high, at 46.33%. It can be seen that the ecological environment of Shilin County has suitable basic conditions. However, with the increase in agricultural activities, a large amount of disorderly burning of straw and excessive use of pesticides, fertilizers and mulch will lead to sudden pollution and environmental deterioration.

(6) In the socio-economic carrying capacity subsystem, the energy consumption of 10,000 yuan GDP has the greatest weight. Shilin County's industrial level is not high—compared with other cities, there is a significant gap, still in the transformation stage of economic growth. Agricultural productivity is not strong, and the level of agricultural modernization is insufficient. Shilin's government report states [52] that Shilin's economic development is not dynamic enough and the quality is not high. The industrial structure is not excellent, and the development transition is slow, with an insufficient level of industrial modernization. This corresponds with the results of the analysis of this paper.

*5.2. Enhancement Strategies*

In order to improve the carrying capacity of resources and the environment in Shilin County; enable the economy, resources and the environment to develop in a coordinated

manner; and promote the sustainable development of Shilin County, the following suggestions are made for the existing problems.

(1)   Promote the efficient and intensive use of soil and water resources and enhance the carrying capacity of soil and water resources. Based on the results of the above analysis, the per capita arable land area and per capita built-up area in Shilin County are more than adequate. However, the urban built-up area is less and the rural built-up area is more, resulting in an imbalance of urban and rural settlements. In order to change this imbalance between urban and rural settlements, it is recommended to integrate the settlements and vacant land in the region with a scattered layout and small scale, and to strengthen the construction of central villages and central communities. Under the premise of keeping the scale of urban and rural construction land unchanged, the land for rural settlements is reduced and the land for urban construction is increased. On the one hand, this meets the demand for urban construction and improves the rural environment, and on the other hand, it promotes the intensive use of construction land and the improvement of carrying capacity.In response to the high degree of exploitation of arable land and the lack of arable land reserve resources, it is proposed to implement the internal development of arable land by constructing high-standard basic farmland and water-saving irrigation projects, strengthening agricultural infrastructure construction, improving the quality and utilization level of arable land, ensuring stable and increased production and enhancing the carrying capacity of arable land.

(2)   Establish environmental awareness, strengthen governance and enhance the ecological carrying capacity. Shilin County uses a large amount of chemical fertilizer, which is prone to causing source pollution in agriculture. It is recommended to promote green agricultural production, use organic fertilizers instead of chemical fertilizers and film pollution prevention to reduce the use of chemical fertilizers and pesticides.Based on the opportunity of conversion of new and old dynamic energy, further adjust the industrial structure and optimize the energy structure. Continue to promote green and low-carbon industrial development, build a multi-level circular economy industry chain and strive to create a national circular economy demonstration county.Carry out in-depth greening actions throughout the region, strengthen ecological construction and enhance the carrying capacity of the ecological environment.

(3)   Strengthen the transformation and upgrading, focus on improving quality and efficiency and promote the steady improvement of socio-economic carrying capacity. Adhere to the main line of supply-side reform and accelerate industrial transformation and upgrading. Promote traditional enterprises to advance to the middle and high end, and focus on building an innovation-driven industrial pattern.Accelerate the transformation of agricultural development, focus on the development of exquisite agriculture and take a modern agricultural development path of high output efficiency, product safety, resource conservation and environmental friendliness.Advance the development of exquisite tourism, using the unique karst landscape to create a leisure tourism industry rich in ethnic characteristics and promote the socio-economic carrying capacity.

## 6. Research Conclusions and Perspectives

Under the current pattern of actively promoting the coordinated development of social regions and the sustainable development of cities and towns, a study on the evaluation of the resource and environmental carrying capacity of Shilin County was conducted as an example. At the regional level, it can provide a reference for the evaluation of the carrying capacity of karst areas. At the national level, it is a complementary improvement to the evaluation of the comprehensive carrying capacity of resources and environments of different landscape types such as mountains, plateaus and hills.

*6.1. Research Conclusions*

(1) The IFMOP model does not consider the economic and energy supply and demand balance within the study region. In this paper, when analyzing the socio-economic carrying capacity, the indicators of the interaction between economy and energy consumption, such as energy consumption per 10,000 yuan GDP and electricity consumption per 10,000 yuan GDP, are added to the calculation model. It is proposed to change the economic development strategy to allow the coordinated development of socio-economy and ecological environment.

(2) The TEF and TEC models consider only ecological impacts and are unable to evaluate indicators of uncertainty and multi-objectivity. The fuzzy integrated evaluation method used in this paper analyzes all economic, ecological and soil and water resource aspects. It makes up for the shortcomings of a single evaluation index and unrepresentative indexes. It can be used as a basis for formulating development strategies in the study area through various aspects.

(3) The linear causality of the DPSIR model oversimplifies the actual situation. The fuzzy integrated evaluation method used in this paper includes both multi-objective and uncertainty objectives in the actual situation as evaluation indicators. The calculation results are more appropriate for the actual situation and provide a credible basis for proposing development strategies for the study area.

(4) Using the fuzzy comprehensive evaluation method, we obtained a comprehensive evaluation score of 0.5969 for the resource carrying capacity of Shilin County, which is at a medium level. The scores for the carrying capacity of soil and water resources, ecological and environmental carrying capacity, and socio-economic carrying capacity are reduced in order.

(5) Through the short board element analysis, we learned that Shilin County currently has no short board elements. However, the land utilization rate is in Early Warning Status. There is an imbalance in the structure of urban land area and rural land area.

(6) Promote the efficient and intensive use of water and soil resources, establish environmental awareness, strengthen ecological management and promote the transformation and upgrading of industrial structure and other measures to comprehensively improve the carrying capacity.

*6.2. Perspectives*

This paper takes Shilin County as the research unit and combines the analysis of shortcoming elements to evaluate the bearing capacity of resources and the environment. Due to factors such as the difficulty of collecting data from county-level units, the paper has some limitations that need to be improved in the future.

(1) Increase the longitudinal comparison on the time scale to reflect the trend of temporal changes and improve the study of resource and environmental carrying capacity on time and space series.

(2) Refine the evaluation unit, take townships as the research object, combine with GIS technology and analyze the variability of resource and environmental carrying capacity among different townships in Shilin County in terms of spatial territory.

**Author Contributions:** All authors contributed to the completion of this article. Material preparation, data collection and analysis were performed by X.M. and J.C. The first draft of the manuscript was written by X.M., and all authors commented on previous versions of the manuscript. Method, funding acquisition and supervision were performed by Y.W. All authors have read and agreed to the published version of the manuscript.

**Funding:** This study was supported by the Natural Science foundation of Guangdong Province of China (2022A1515011200), the Science and Technology Planning Project of Guangdong Province of China (STKJ2021129) and the State Key Laboratory for Geo-Mechanics and Deep Underground Engineering of China University of Mining & Technology (SKLGDUEK2005).

**Informed Consent Statement:** Not applicable.

**Data Availability Statement:** The datasets generated during and/or analyzed during the current study are available from the corresponding author on reasonable request.

**Acknowledgments:** The authors are grateful to two anonymous reviewers for their valuable comments and constructive suggestions. We also thank Zhengguo Song and Shuilong Shen in Shantou University for their guiding and reviewing.

**Conflicts of Interest:** The authors declare that they have no known competing financial interests or personal relationships that could have appeared to influence the work reported in this paper.

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
