# Peer review of "Resource Carrying Capacity Evaluation Based on Fuzzy Evaluation: Validation Using Karst Landscape Region in Southwest China"

_sustainability, doi:10.3390/su142416548_

Round 1
Reviewer 1 Report
Decision: Major Revision
Summary
I think that it is important to have a comprehensive, integrated, and quantitative method of analysis for multiple policy indicators. However, there are many similar studies. Among them, it is necessary to clarify the characteristics of this research.
The parts of the methodology are written unclear. In particular, it is not clear from this paper why you used fuzzy method, it is necessary to explain clearer reason.
If the methodology is not properly explained, I could not determine the validity of this study.
Each part
Figure.1
Please add Reference.
Line52-65
I do not understand reason why you used Fazzy method. Please describe the characteristics of this method and why you used it.
Figure.2
In this figure, it appeared that both the Delphi method and the AHP method are used. However, it is not clear from the text that the Delphi method is used. Please change the Figure to match the text.
Line89-139
Please add Reference for basic information of target area.
Figure.5
I don't understand the need to make it a radar chart. Please make it Table form and clearly indicate the numbers and area shares.
Table.2 & 3
It is described in Line 149. Please reposition these tables that it is immediately after this text.
Line177 etc
The formula text is in bold, as in DI. Please change to italics.
Section 2.3
From the text it does not appear that the Delphi method was used. Also, it is important to know who and what questions were asked when the AHP was conducted. Please provide a description this information.
Table 5
It is the same as Bruce et. al. (1989), please provide a citation if correct this.
I think that the original paper was used n, not Rank, and we could not use Rank1 and Rank2, where RI is 0. Then, I think that this table number for Ranks.
Please add an explanation.
Equation (19)
Please add an explanation for di.
Did this formula represent a 5-step fuzzy function? The significance of using this method is unclear because you have not specified reason of using the fuzzy function. Please add an explanation.
Line346-349, Table 9
This text should be moved to Methodology part.
Author Response
On behalf of all the authors, we thank you very much for giving us an opportunity to revise our manuscript. We appreciate editors and reviewers very much for your positive and constructive comments and suggestions on our manuscript entitled " Resource carrying capacity evaluation based on fuzzy evaluation: Validation using karst landscape region in southwest China ".
We revised the manuscript in accordance with the reviewer’s comments, and carefully proofread it to modify the inaccurate expression and minimize the similar errors that may arise. Attached is our responses to the comments.

Reviewer 2 Report
The analysis of this article is relatively simple, only doing resource carrying capacity evaluation based on fuzzy evaluation, nothing but change a study area, Shilin County, karst landscape region in southwest China, lack of in-depth research.
(1) In the introduction, the summary and review of the existing literature is relatively simple, and no commentary is made. The author does not explain clearly the significance of evaluating resource carrying capacity, nor do illustrate why the study of resource carrying capacity evaluation in Shilin County was done? The author does not introduce the scholarly contributions of this article.
(2) It would be wonderful if the shortcomings and gaps in the literature were clarified, particularly with regard to how the proposed strategy intends to fill up the gaps in the literature.
(3) Construction of evaluation index system lacks theoretical basis and literature support. The author uses a lot of words to introduce the method, which is very simple and not different from the traditional method.
(4) The empirical results were not compared to with the empirical literature.
(5) There is a need to improve the analysis of outcomes based on the contributions of the manuscript.
Author Response

(The authors gave the same response as above.)

Round 2
Reviewer 1 Report
Decision:Major Revision
Summary
I think that developed the data-driven system as a new method of adaptive control of DHC very interesting study.
However, there is lack of explanations in an important part of the manuscript of the research, which should be improved.
Each part
Line45
"District Heating system in Bronderslev, Denmark" may not be known to the reader. Please provide references and explain the target are of your research.
Equations
There is an equation like in equation (3) where the right and left sides are written by Programing type grammar. In this case, the equations are not equal and should be corrected for each equation.
2.2.3
It is concerned with setting the demand value for this study calculation. Please add an explanation of this part of the projection. In particular, please clarify how the demand is estimated.
Fig.5
Please make the text larger.
This simulation results show a downward overshoot in estimation lines, please provide an explanation for these in the manuscript.
Fig.6-10
Please make the text larger.
Please add more detailed explanations in the manuscript of how this figure should be read so that the reader could understand the meaning of each legend means.

Author Response
Dear Editor and Reviewers,
On behalf of all the authors, we thank you very much for giving us an opportunity to revise our manuscript. We appreciate editors and reviewers very much for your positive and constructive comments and suggestions on our manuscript entitled " Resource carrying capacity evaluation based on fuzzy evaluation: Validation using karst landscape region in southwest China ".
We revised the manuscript in accordance with the reviewer’s comments, and carefully proofread it to modify the inaccurate expression and minimize the similar errors that may arise. Here below are our responses to the comments.

Reviewer 2 Report
(1)The summary of the shortcomings and gaps in the literature were not clarified, particularly with regard to how the proposed strategy intends to fill up the gaps in the literature. The current revision is not enough. Further modifications were required.
(2) The empirical results were not compared to with the empirical literature. The current revision did not address this issue.
Author Response

(The authors gave the same response as above.)
